# Fabrication of Sulfated Heterosaccharide/Poly (Vinyl Alcohol) Hydrogel Nanocomposite for Application as Wound Healing Dressing

**DOI:** 10.3390/molecules27061801

**Published:** 2022-03-10

**Authors:** Yang Liu, Ning Wu, Lihua Geng, Yang Yue, Quanbin Zhang, Jing Wang

**Affiliations:** 1College of Materials Science and Engineering, Qingdao University, 308 Ningxia Road, Qingdao 266003, China; 2019020477@qdu.edu.cn; 2CAS and Shandong Province Key Laboratory of Experimental Marine Biology, Center for Ocean Mega-Science, Institute of Oceanology, Chinese Academy of Sciences, 7 Nanhai Road, Qingdao 266071, China; wuning@qdio.ac.cn (N.W.); lhgeng@qdio.ac.cn (L.G.); yueyang@qdio.ac.cn (Y.Y.); qbzhang@qdio.ac.cn (Q.Z.); 3Laboratory for Marine Biology and Biotechnology, Qingdao National Laboratory for Marine Science and Technology, Wenhai Road, Aoshanwei, Jimo, Qingdao 266237, China

**Keywords:** UF/PVA hydrogel nanocomposite, sulfated heterosaccharide, wound healing, inflammatory regulation

## Abstract

Nowadays, natural polysaccharides-based hydrogels have achieved promising results as dressings to promote skin healing. In the present study, we prepared a novel hydrogel nanocomposite with poly(vinyl alcohol) (PVA) and sulfated heterosaccharide (UF), named UPH. The SEM results showed that the UPH had dense porous structures with a high porosity and a specific surface area. The UPH had a good swelling property, which can effectively adsorb exudate and keep the wound moist. The in vitro experiments results showed that the UPH was non-cytotoxic and could regulate the inflammatory response and promote the migration of fibroblasts significantly. The phenotypic, histochemistry, and Western blot analyses showed UPH treatment accelerated the wound healing and recovery of skin tissue at wound sites in a C57BL/6 mouse model. Furthermore, the UPH could promote the inflammation process to onset earlier and last shorter than that in a normal process. Given its migration-promoting ability and physicochemical properties, the UPH may provide an effective application for the treatment and management of skin wounds.

## 1. Introduction

The skin is the largest organ by the surface area in the human body. It protects internal tissue from mechanical damage, microbial infection, ultraviolet radiation, and extreme temperatures [1,2,3]. The process of wound healing mainly includes the following four stages: haemostasis, inflammation, proliferation, as well as migration and maturation [3]. These stages involve systematic actions of various cells and the fluctuation of cytokine levels [1]. The inflammation stage is accompanied by releasing chemokines and recruiting inflammatory cells, which are mainly neutrophils and macrophages. Macrophages can express tumor necrosis factor-α (TNF-α) and interleukin-6 (IL-6) during the early stage of the inflammation, which are crucial in wound healing [4]. Many studies have focused on the inflammatory regulation and the macrophage phenotypes expressed during wound healing following tissue injury [5,6].

The concept of moist wound healing has been paid more and more attention to in nursing traumatic patients since 1962 [7]. Keeping the wound area moist can be conducive to the micro-environment of wound healing, which is moderately moist, low oxygen, slightly acidic, clean surrounding environment, to accelerate the wound healing [8]. Hydrogels, foams, and films are used as biomaterials for wound healing and have good water absorption to keep the wound area moist [9,10,11]. The hydrogel has a three-dimensional hydrophilic polymer network connected by the covalent bond, hydrogen bond, van der Waals interactions, or other physical or chemical bonds [12]. It is an ideal material for wound dressing, because the hydrogel can absorb excessive water/exudate, maintain a moist environment, promote gas exchange, adapt to all kinds of wounds and reduce pain and inflammation through the release of bioactive compounds [13].

The functional properties of natural polysaccharides (such as fucoidan, alginate, and gellan gum) can be made of hydrogel dressings and used in the wound healing process [14]. Fucoidan is a sulfated polysaccharide mainly made of fucose and sulfated group, extracted from brown seaweeds. Fucoidan as a pharmaceutical ingredient is incorporated in a hydrogel to enhance the gelling for the hydrogel and accelerate the wound healing [14]. Karuppusamy et al. prepared and evaluated a fucoidan/alginate-polyethylene glycol-gellan gum hydrogel in an ICR mouse model, and the results found the hydrogel combined with low-level laser therapy could promote wound healing and reduce scars [15]. Kaoru et al. prepared a hydrogel sheet composed of a blended powder of alginate, chitin/chitosan, and fucoidan and evaluated the wound healing activity. They found the hydrogel could maintain the reactive oxygen species (ROS) level and alleviate the cell damage caused by oxidative stress and free radicals [16].

Sulfated heterosaccharide (UF) is a polysaccharide extracted and purified from brown seaweeds with high uronic acid, low fucose, and sulfate group contents. Our previous study found UF exhibits various biological properties, including antioxidant, neuro-protective, and anti-apoptotic ones [17,18,19]. Among the samples purified from the brown seaweeds, UF with the highest uronic acid content and the lowest fucose and sulfated groups contents had the strongest neuroprotective effect both in vitro and in vivo. UF could increase the level of antioxidant enzymes and reduce the level of lipid peroxidation in Parkinson’s disease (PD) mice. UF is another kind of poslysaccharide extracted from brown seaweeds, which has a similar monosaccharide composition and antioxidant activity to fucoidan. We supposed UF can be used in wounding healing. However, there is a lack of comprehensive studies on wound healing hydrogels based on UF.

The aim of this study is to prepare a UF/poly(vinyl alcohol) (PVA) hydrogel nanocomposite (UPH) with UF as the main active ingredient and evaluate the wound healing activities in vitro and in vivo. This study provides an experimental foundation and a theoretical basis for the application of UF in wound healing therapy and provides a scientific basis for the development of new and effective wound healing dressing.

## 2. Results

### 2.1. The Composition of the UF

The chemical composition (%, dry weight) of UF is shown in Table 1. The main chemical components were fucose, uronic acid, and sulfate groups. Fucose, galactose, and mannose were the primary monosaccharides, which occupied more than 80% in the neutral sugar. The molecular weight of UF was 7655 Da.

### 2.2. The Structure of the UPH

The color of the UPH was different from that of PVA hydrogel. We assumed the UPH was loaded with UF. To evaluate the stability of the UPH, the morphology of the UPH was observed in deionized water (DW) (Figure 1b). We found in DW, the UPH could remain morphologically unchanged for a long time.

The SEM analysis (Figure 1c) revealed the UPH had a high specific surface area and a dense porous structure. The diameters of pores ranged from hundreds of nanometers to several micrometers. Meanwhile, some small particles with a size of hundreds of nanometers could be observed attached to the skeleton of PVA, which were thought to be combined with UF.

As shown in Figure 1d, the UPH had characteristic peaks of both UF and PVA. The peaks located at 1648 and 1236 cm^−1^ were derived from the carboxyl and sulfate groups in UF.

### 2.3. Swelling Ability and Gel Content

The microstructure of the UPH guaranteed its breathability and high water-absorption ability. As shown in Figure 2a, the swelling experiment found the UPH had a high swelling rate, which means it could absorb wound exudate well, thereby providing a moist healing environment. On the other hand, the UPH had little mass loss after swelling and drying (Figure 2b). The result indicated that the UPH had a stable structure and the reduced quality was mainly the dissociation of UF.

### 2.4. The Dissociations of UF at Different pH Values

We tested the dissociations of UF from the UPH in different pH solutions. The results found the release patterns in the three different pH solutions (pH 7.4, pH 5.6, and pH 2.2) were similar. In all of the solutions the release rate of UF at the first 8 h was rapidly and then slowed down (Figure 2c). The results indicated that the dissociation of UF from the UPH was very stable and could not affected by the pH change in wound areas.

### 2.5. The Biocompatibility, Pro-Inflammatory, and Migration Activity

The cytocompatibility of the UPH was further evaluated by a standard 3-(4,5-dimethylthiazol-2-yl)-2,5-diphenyltetrazolium bromide (MTT) assay using L929 cells (Figure 3a). The results showed that compared with the normal control group the cell viability of L929 cells was not affect by PVA and the UPH and more than 90% of the cells survived in the hydrogel extracts. The results indicated that UPH had no significant cytotoxicity in L929 cells. The blood compatibility of UPH was investigated by a hemolysis assay. The hemolysis rates of the UPH and PVA were much lower than 0.5% (Figure 3b), which means the UPH had no obvious hemolysis activity. These results demonstrated that the UPH had excellent biocompatibility and little cytotoxicity, and it could be used in wound healing applications without causing long-term effects on biological growth.

The pro-inflammatory activity of the UPH was determined by the change of the nitrite (NO) concentration in the supernatant of a RAW264.7 cell medium (Figure 3c). The concentration of NO in the UPH-treated cells was significantly higher than in the normal control cells. However, the concentration of NO in the PVA group was basically the same as that in the normal control group. These results indicated that the UPH had an obvious pro-inflammatory activity.

Cell migration plays an important role in the wound healing process. In this study, a transwell assay was performed to investigate the promoting effect of the UPH on the migration of L929 cells (the migrated cells were stained purple). As the results shown in Figure 4a,b, after treated with the UPH, the number of the cells migrated to the lower chamber were significantly increased than those of the cells treated with a complete medium (CM).

### 2.6. Wound Healing in a C57BL/6 Mouse Model

To evaluate the effects of UPH on wound healing, we generated full-thickness cutaneous wounds on the back of C57BL/6 mice. The wound surfaces were cleaned and changed dressing once a day. During the experiment, none of the wound surfaces had apparent infection at any stage after surgery (Figure 5a). On the first day after injury, the wound area in each group expanded, which might be the result of the combined effects of skin contraction caused by pain stimulation and mechanical damage.

As shown in Figure 5a, at day 3, the wound area of the UPH-treated mice was smaller than that of the gauze control group, which was similar to that of the recombinant human epidermal growth factor gel (yeast) (rhEGF)-treated group. At day 7, only a little wounds of the mouse treated with rhEGF and the UPH remained. At day 11, the wound of the UPH-treated mice was totally healed. The results showed that the wound-healing rate of the UPH group was faster than that of the gauze control group, and UF played a positive role in wound healing.

The histological analyses of wound re-epithelialization were performed at days 3, 7, and 11 after surgery to assess the extent of wound healing and regeneration. The H&E staining showed the wound healing and re-epithelialization in the rhEGF- and UPH-treated groups were higher than in the the gauze control group at days 3, 7, and 11. On the 3rd, 7th, and 11th days, the degrees of re-epithelization in the UPH-treated group were exceeded 35%, 50%, and 90%, respectively, which were similar to those in the rhEGF-treated group, and at days 3 and 7 the UPH-treated group was even better than the rhEGF-treated group (Figure 6a).

On the 11th day, the scars of the mice in the UPH-treated group were exfoliated, and the epidermis grew completely. The epidermis under the cuticular layer of the mice in rhEGF-treated group grew completely, but some scars did not fall off, while the wounds of the mice in the gauze control group were not completely closed.

Collagen deposition and maturation were further assessed in each group by Masson’s trichrome (MT) staining (Figure 6b). As shown by light blue staining, the UPH-treated group not only had a higher collagen deposition, but also exhibited a more orderly collagen arrangement than the other groups at day 11.

Western blotting was used to examine the expression of inflammatory factors in wound tissue at days 3 and 7 after surgery (Figure 7). The expression of monocyte chemoattractant protein-1 (MCP-1) in the UPH-treated group was significantly increased at day 3 and was reduced at day 7. The expression of IL-6 in the UPH-treated group was basically unchanged, while the expression of TNF-α in the UPH-treated group had a decreasing trend.

## 3. Discussion

In the present study, we extracted and purified UF from *Saccharina japonica* (Laminariaceae). Compared with some animal-derived pharmaceutical ingredients [20,21], the UF extracted from Laminariaceae had a broader source. A novel UPH was prepared using UF and PVA under cyclic freeze-thaw conditions. The differences in the color and characterization between the UPH and PVA hydrogel indicated that UF was connected to the PVA three-dimensional skeleton. The UPH had a good elasticity (Appendix A) and an easily skin-fit ability. It could absorb exudate from the wound and maintain a moist wound-healing environment because of its high water-absorption properties. These characteristics were consistent with the moist wound healing theory [22]. The different stages of the wound healing process were accompanied by changes in the wound exudate in terms of pH [23]. The UPH could stably release UF without being affected by pH, which could ensure that it takes effect in all stages of wound healing.

During the inflammation stage of wound healing, macrophages were recruited to the damaged skin site to implement a pro-inflammatory activity [24]. The cytology test results found that the UPH was able to significantly increase the secretion of NO in RAW264.7 cells, which indicated that UPH could regulate inflammation in vitro. Furthermore, NO can regulate inflammation and stimulate angiogenesis and cell proliferation [25]. Inflammatory factors play an important role in the wound healing process. The Western blot results showed the changes of inflammatory factors in each group of mice. Among inflammatory factors, MCP-1 changed most obviously on the third and seventh days. Some studies have shown that MCP-1 plays a critical role in monocyte recruitment into sites of immune responses [26]. We inferred that the UPH could promote the recruitment of monocytes/macrophages. Combined with the changes in wound conditions, we inferred that the UPH could promote wound inflammation, but would not lead to prolonged inflammation time, and might even shorten the inflammation time.

The proliferation stage is accompanied by the formation of granulation tissue and the deposition of collagen fibers [27]. Fibroblasts are ubiquitously present in the connective tissue of every organ system where they deposit and remodel the extracellular matrix (ECM) [28]. UPH could promote the migration of fibroblasts. We supposed that UPH could promote the cell migration during the proliferation stage. The MT staining found that the deposition of collagen fibers in the UPH-treated group was much better than that in the gauze control group. The results confirmed that UPH could promote the granulation tissue formation and collagen fibers deposition in the proliferation stage.

The external application of rhEGF could promote the synthesis of DNA, RNA, and hydroxyproline during wound tissue repair and could accelerate the generation of wound granulation tissue and epithelial cell proliferation, thus shortening the wound healing time. The healing rate of the rhEGF-treated group was faster than that of gauze control group, but the wound skin tissue remodeling was worse than that of the UPH-treated group on the 11th day, which was consistent with the report of Wu et al. [21]. The rhEGF-treated group mice had higher IL-6 levels on the seventh day. We inferred that exogenous growth factors may cause the disorder of some cytokines in the mouse wound. For the UPH, there was no such anomalous trend.

Wu et al. purified and clarified snail glycosaminoglycan (SGAG) from a cultured China white jade snail [21]. Pharmacological experiments showed that the wound healing of diabetic mice in the SGAG-H group was accelerated, and its healing curve overlapped greatly with that in the rhEGF group, which was similar to that of the UPH-treated group. However, in comparison, the source of UF was more extensive.

## 4. Materials and Methods

### 4.1. Materials

*Saccharina japonica* (Laminariaceae), cultured at the coast of Rongcheng, China, was collected in September 2020, authenticated by Prof. Lanping Ding and stored as a voucher specimen (No. 126) in the Jerbarium of the Algal Chemistry Department, Institute of Oceanology, CAS. The fresh algae were promptly washed, dried at 40 °C, powdered using a grinder (Beijing Zhongxing Weiye Century Instrument Co., Ltd., Beijing, China, FW-100) and kept in sample bags at room temperature for use. We purchased PVA from Shanghai Macklin Biochemical Technology Co., Ltd., Shanghai, China, All the chemicals used were of analytical grade.

### 4.2. Preparation and Analysis of UF

Fucoidan was extracted from *Saccharina japonica* according to the method of Wang et al. with the reaction time increased from 3 to 4 h [19]. Then, low-molecular-weight fucoidan (LMWF) was prepared using ascorbate and hydrogen peroxide (30 mmol/L; *v*:*v*, 1:1). After reaction for 2 h, the solution was dialyzed using dialysis membranes with a cutoff value of 3600 Da and precipitated with ethanol. After lyophilization, UF was further purified using DEAE-Sepharose FF exchange chromatography with 0.5 M NaCl eluted.

The flow chart is shown in Figure 1.

The total sugar content was determined by the phenol sulfate method [19]. The fucose content was determined by the cysteine hydrochloride-sulfuric acid method with L-fucose as the standard [29]. The sulfate content was analyzed by the ion chromatograph with potassium sulphate as the standard [30]. The uronic acid content was measured using modified carbazole with D-glucuronic acid as the standard [31]. The neutral sugar composition and the molecular weight were determined by high-performance liquid chromatography with a Photo-Diode Array (PDA) detector, C18 column and Refractive Index Detector (RID) detector, and a TSK G3000 PW × 1 gel column [32].

### 4.3. Preparation of the UPH

A UPH was prepared using the method of Yao et al. [33]. UF was added into a 10% PVA aqueous solution, with the same volume ratio of PVA:UF = 20:1, and stirred at 90 °C and 200 rpm to complete dissolution. The mixed solution was added to a mold and placed at −20 °C for 12 h and then at room temperature for 12 h after 5 cycles [34].

### 4.4. Characterization of the UPH

#### 4.4.1. SEM and FTIR Analysis

SEM analysis was performed using a Quanta 400F Field Emission scanning electron microscope. The freeze-dried UPH sample was cut vertically, and its cross-section surfaces were coated with gold and then were observed.

FTIR spectra were obtained on a NicoletiS10 FTIR spectrometer in the region from 4000 to 400 cm^−1^. UF powder, PVA, and UPH flakes were used for analysis.

#### 4.4.2. Swelling Rate and Gel Content Analysis

The lyophilized UPH was weighed and recorded as M0. Then, soak it was soaked in DW for 1d, taken out, immersed in DW from the surface, weighed and denoted as m1 [35]. The swelling rate of UPH was described as:$$\mathrm{swelling}\;\mathrm{rate}\;\left(\%\right)=\frac{m1-m0}{m0}\times 100\%.$$

The expanded sample was freeze-dried again and weighed, denoted as m2. The swelling rate of the expanded sample was expressed as:$$\mathrm{gel}\;\mathrm{content}\;\left(\%\right)=\frac{m2}{m0}\times 100\%.$$

#### 4.4.3. The Dissociation of UF from the UPH

The dissociation of UF was performed in aqueous solutions at various pH values that were adjusted using HCl (0.5 M) or NaOH (0.5 M) at 37 °C. UPH was immersed in a certain amount of normal saline (NS) according to relevant standards, and the pH values were adjusted to 7.4, 5.6, and 2.2 separately [36]. Since the fucose content in UF can be obtained in the component analysis, we measured the concentration of fucose to estimate the dissociation of UF. The concentrations of fucose in the solutions were determined by the cysteine hydrochloride-sulfuric acid method with L-fucose as the standard.

### 4.5. Cytocompatibility and Hemocompatibility Investigation

The cytocompatibility and hemocompatibility were investigated to evaluate the biocompatibility of UPH. Because the hydrogel preparation process cannot maintain completely sterile conditions, we used the hydrogel extracts to simulate the state of the medium after the hydrogel was added. The hydrogel extracts were prepared as following: 500 mg UPH was soaked in 25 mL Dulbecco’s Modified Eagle Medium (DMEM) for 24 h, after filtering with a 0.2 μm filter membrane, and then 10% fetal bovine serum was added into the solution.

The cytocompatibility of UPH was measured using an MTT assay (Sigma-Aldrich, St. Louis, MO, USA) according to the manufacturer’s instructions. L929 fibroblasts cells (5.0 × 10^3^ cells/well; Shanghai Cell Bank, CAS) were vaccinated to a 96-hole plate and incubated for 24 h, at 37 °C under 5% CO_2_ atmosphere. The normal control group was further cultured with a CM, while the experimental group cultured with hydrogel extracts. The MTT test was performed after continuous culture for 24 h, and the absorbance was measured at 490 nm with a microplate analyzer.

The hemocompatibility of the UPH was investigated using a hemolysis assay.

A 2% red blood cells suspension was prepared using fresh whole blood from SD rat. Then, 2.5 mL red blood cell suspension were added into a 10 mL test tube. UPH and the PVA hydrogel with a similar shape and mass were added separately, After that, normal saline (NS) was added, until the total volume was 5 mL. DW was added to the positive control group, and NS was added to the negative control group without samples. The test tube was placed in a 37 °C thermostat for 1 h. Then, the suspensions were centrifuged, and the supernatant was taken out for analysis. The absorbance was determined at 540 nm, and the hemolysis rate was calculated as following:$$\mathrm{Hemolysis}\;\mathrm{rate}\left(\%\right)=\frac{\left(\mathrm{ODh}-\mathrm{ODn}\right)}{\left(\mathrm{ODp}-\mathrm{ODn}\right)}\times 100\%,$$
where ODh is the supernatant absorbance of the experimental group, ODp is the supernatant absorbance of the positive control group, and ODn is the supernatant absorbance of the negative control group.

### 4.6. Cell Migration Assay

In the cell migration experiment, a 24-well cell culture plate with a transwell chamber was used. The L929 fibroblast cells in the logarithmic growth phase were suspended in a serum-free medium. A transwell chamber was added with 100 μL serum-free media and 200 μL cell suspension (cell count: 2.5 × 10^5^ /mL). The lower wells were added into 750 μL serum-free media, CM, and hydrogel extracts in the normal control group and experimental groups separately. The cells maintained at 37 °C and 5% CO_2_ for 16 h. Then the transwell chamber was taken out and washed with PBS twice; then it was fixed with 4% paraformaldehyde at room temperature for 30 min, washed twice with PBS, permeated with 100% methanol and placed at room temperature for 20 min. After that, the transwell chamber was washed with PBS twice, stained with 0.1% crystal violet at room temperature for 20 min without light, discarding crystal violet dye, washed with PBS for 3 time. The upper layer of unmigrated cells was gently erased with cotton swabs and washed with PBS for 3 times. Its pictures were taken under a microscope at 100 times; five non-repeating areas were selected for photography.

### 4.7. The Concentration of NO

RAW 264.7 cells were vaccinated to a 96-hole plate and cultured with hydrogel extracts for 24 h. Nitrite concentration was determined using a NO assay kit (Beyotime Biotechnology). The medium supernatant was transferred (50 μL) and mixed with 50 μL Griess Reagent I and 50 μL Griess Reagent II and incubated at 25 °C for 10 min. The absorbance was measured at 540 nm. A standard curve using NaNO_2_ was constructed to determine NO content in culture fluid.

### 4.8. In Vivo Skin Wound-Healing Experiment

#### 4.8.1. Animals Care and Diet

All animal procedures were carried out in accordance with animal care and approved by the Institute of Science Ethics Committee. C57BL/6 mice were fed in standard polypropylene cages (4 mice per cage) under controlled conditions (23 ± 2 °C, relative humidity of 50% ± 5%, 12:12 h light/dark cycle). All mice were fed a normal pellet diet with ~3100 kcal/kg (5.7% lipids, 22.1% proteins, and 72.2% carbohydrates).

#### 4.8.2. Study Design

C57BL/6 mice (25 to 30 g, 6 weeks of age) were provided by the Jiangsu Jicuiyaokang Biotechnology Co., Ltd. After 2 weeks of sedation, 36 mice were divided into the gauze and two treated groups. A full-thickness excisional skin wound (7 mm in diameter) was made on the backs of mice. The gauze control group used a gauze to cover the wound after normal care, and the treated groups used UPH and rhEGF on the wound after normal care. The wound was checked and cleaned up, and the dressing was changed every day.

The wound area was recorded by a digital camera at days 0, 1, 3, 7, and 11. We used the wound residual area (%) to characterize the degree of wound closure in mice:$$\mathrm{wound}\;\mathrm{residual}\;\mathrm{area}\;\left(\%\right)=\frac{\mathrm{Sn}}{S0}\times 100\%,$$
where S0 is the initial wound area, and Sn is the wound area at different time points [14].

#### 4.8.3. Histological Analysis

The wound tissue was taken at 3, 7, and 11 days after the operation and fixed in a 4% paraformaldehyde solution. The samples were embedded in paraffin for routine histological processing, and then tissue sections of about 5 μm were prepared for histological examination. The sections were stained with H&E and MT.

#### 4.8.4. Western Blot Analysis

Skin protein was extracted to perform western blot. Wound tissue (20 mg) was rinsed twice with cold PBS and lysed in a lysis buffer (containing 0.1 mM PMSF and a protease inhibitor). Protein concentration was measured by a BCA kit (Solaibao Technology Co., Ltd., Beijing, China). Proteins were separated by polyacrylamide gel electrophoresis (SDS-PAGE), transferred to a PVDF membrane, blotted with each antibody and detected by using an ECL reagent. TNF-α (sc-52746), MCP-1 (sc-52701), and IL-6 (sc-57315) were measured.

### 4.9. Statistical Analysis

GraphPad Prism 8.0.2 software was used for statistical analysis. The values are shown as the mean ± SD. Data were analyzed using one-way analysis of variance followed by Dunnett’s post hoc test for multiple comparisons. *p*-values of <0.05 were considered to indicate statistical significance.

## 5. Conclusions

A novel UPH was prepared, and the wound-healing activity was evaluated both in vitro and in vivo. The UPH had a good elasticity and an easily skin fit ability, and it could absorb exudate and release UF stably without being affected by the pH changes in the wound area. The UPH could accelerate the wound healing process by regulated inflammation and proliferation stages. It could promote the recruitment of macrophages in the early stage of inflammation and end the inflammation stage quickly. The UPH not only was beneficial to wound healing directly, but also promoted tissue remodeling significantly. The application of UF in dressings can broaden the exploration of sulfated polysaccharide in the field of biological activity. In addition, as a kelp-derived polysaccharide, its cost and yield are guaranteed to a certain extent. This study confirmed that UF as the main active ingredient could be used on wound healing therapy, which provides a scientific basis for the development of new and effective wound healing dressings.

## Data Availability

Not applicable.

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
