# Peer review of "Fabrication of Sulfated Heterosaccharide/Poly (Vinyl Alcohol) Hydrogel Nanocomposite for Application as Wound Healing Dressing"

_molecules, 2022, doi:10.3390/molecules27061801_

Round 1
Reviewer 1 Report
Manuscript: molecules-1568117
Title: Fabrication of sulfated heterosaccharide/poly (vinyl alcohol) hydrogel nanocomposite for application as wound healing dressing
Review report:
This study aimed to prepare a new hydrogel based on sulfated heterosaccharide/poly(vinyl alcohol) for application as wound healing dressing material. The manuscript is good, covering the essential in vitro and vivo tests regarding the subject. The study was well designed and the results were promising. Some parts like discussion can be improved to show the cause and strength of the findings. Also, comparison with some other materials reported in the filed may be advantage of the study. The manuscript is suitable and can be accepted.
Here are some comments for the authors.
Title: Please use poly(vinyl alcohol) instead of poly (vinyl alcohol). Remove the space between poly and bracket (also check it in the whole manuscript).
Abstract: Line 15: add UF/PVA composition ratio.
Introduction:
- Line 51: compounds.
- Authors need to define abbreviations at the first time they appear in the manuscript, e.g., line 86: S. japonica.
- Lines 71-73: Make the point clearer.
- Line 72: correct poslysaccharide to polysaccharide.
Results:
- line 86: S. japonica (italic).
- Line 160: space after (a).
- Figure 5b, Figure 6b and d, and Figure 7 b and d: higher resolution of graphs is recommended.
Materials and methods:
- Formatting: words in titles of section, subsection, and subsubsection started with capital letters.
- Line 248: As you mintioned minor modification of the extraction method, it is better to define it even in brief.
- Line 251: Is it necessary to write F as capital in “Fucoidan”? Also, you may add coma after fucoidan instead of period.
- Line 257: please check if the cited reference is the relevant one.
- Line 262: What do you mean by respectively? You may need to mention the method for molecular weight determination as well.
- Line 291: Is it 5 mL or 5 mg?
- Line 323: room. Also, for consistency use min instead of minutes as it is in the next line.
- Line 324: Separate number from unit (20 min).
Patent: Is this section part of the report?
Author Response
Thank you for your hard working on our manuscript. Attachment is our response with the manuscript in amendment mode and you can see all changes from it.

Reviewer 2 Report
This work entitled “Fabrication of sulfated heterosaccharide/poly (vinyl alcohol) hydrogel nanocomposite for application as wound healing dressing” and written by Yang Liu, Ning Wu, Lihua Geng, Yang Yue, Quanbin Zhang and Jing Wang is an article reporting the preparation of hydrogel nanocomposite with Poly (vinyl alcohol) (PVA) and sulfated heterosaccharide (UF), that could accelerate the wound healing process by regulated inflammation and proliferation stages. Moreover, it also promoted tissue remodeling.
For me, some basic characterization of the gel is missing. The authors write that “The color of UPH was different from PVA hydrogel, we thought UPH had been loaded with UF”. Maybe it is a good idea to check the UPH hydrogel and its precursors PVA and UF with techniques like elemental analysis and FTIR. This way they can verify the existence of the UF content in the hydrogel.
The authors have chosen appropriate and up to date references. However, the authors should check again the whole text as the English language needs to be polished. I have spotted some mistakes especially in the materials and methods part.
Some examples:
Page 8, line 240: “vouvher”
Page 8, line 244: “Poly (vinyl alcohol) (PVA) (Shanghai Macklin Biochemical Co., Ltd).” is an incomplete sentence
Page 8, line 244: “After lyophilization to obtain low molecular weight Fucoidan.” is an incomplete sentence
Page 9, lines 275-277: the text needs to be rephrased
The authors need also to check the uniform format of the references.
Author Response

(The authors gave the same response as above.)

Reviewer 3 Report
The problem of obtaining hydrogel nanocomposites for application as wound healing dressing is important and relevant. However, the manuscript needs to be improved. Some comments are: 1.There are typos in the text: Pg 2, line 51 com-pounds; Figure 2. (a) Swelling rate of PVA hydrogel and UPH; (c) The gel content of PVA hydrogel and UPH; (c) The dissociation of UF at different pH values 2. Figure 1 is difficult to understand. The authors need to put more information in the figure legend, and add legend to the photographs. Fig. 1(c) The SEM micrographs of UPH - What are the differences in the two SEM micrographs? 3. Pg 2, line 89. «The color of UPH was different from PVA hydrogel, we thought UPH had been loaded with UF». Pg 7, line 198. «The difference in color and characterization between UPH and PVA hydrogel indicated that UF is connected to the PVA three-dimensional skeleton». It is impossible to draw a conclusion about the inclusion of a substance in the composite only by color. It is necessary to carry out at least FTIR analysis. 4. Pg 7, line 199-200 «The UPH had good elasticity and easily skin fit ability». This is an unfounded statement. A quantitative study of physical and mechanical properties of the composite is needed. 5. Chapter 4 is not well written. It would be necessary to expand the range of methods for studying the physicochemical properties of the composite and add methods for determining the physicomechanical properties. The authors performed only SEM analysis , swelling rate and gel content analysis, and the dissociation of the UF from UPH. More information about the characteristics of the composite could be interested in order to stablish the potential applications of this material. Is it possible sample evaluations using FTIR analysis, NMR spectroscopy, X-ray, strength, tensile analysis and others? 6. In chapter 4.4.3 the authors write that they measured the concentration of UF in the solution. But the description of the method is missing. This needs some explanation.Author Response
Thank you for your hard working on our manuscript. Attachment is our response with the manuscript in amendment mode and you can see all changes from it.

Reviewer 4 Report
Overview: The submitted paper focuses on the fabrication of sulfated heterosaccharide/poly (vinyl alcohol) hydrogel nanocomposite and its application as a wound-healing dressing. The application of new material as wound-healing dressing is still in growing step. So, this research is important to the field. The manuscript is well written and the research well conducted. So, I kindly suggest the publication of this study. Please, check the minor points to address:
Authors should compare their work with other published work. What is the significance of their work with previous work?
https://www.mdpi.com/1422-0067/21/22/8831/htm
https://www.mdpi.com/1999-4923/13/10/1666/htm
Author Response

(The authors gave the same response as above.)

Round 2
Reviewer 2 Report
I suggest that the revised manuscript gets one more time scanned, as far as the English language is concerned, and then it is ready to be published in Molecules in its present form.
Reviewer 3 Report
The authors have adequately addressed all of my previous comments. They authors have taken into account corrections and recommendations from the reviewer.